# Comment on "Can assimilation of crowdsourced data in hydrological modelling improve flood prediction?" by Mazzoleni et al. (2017)

Daniele P. Viero[1]

[1]Department of Civil, Environmental, and Architectural Engineering, University of Padova, via Loredan 20, 35131, Padova (Italy).

*Correspondence to:* Daniele P. Viero (daniele.viero@unipd.it)

**Abstract.** Citizen science and crowdsourcing are gaining increasing attention among hydrologists. In a recent contribution, Mazzoleni et al. (2017) investigated the integration of crowdsourced data (CSD) in hydrological models to improve the accuracy of real-time flood forecasts. The Authors used synthetic CSD (i.e., not actually measured), because real CSD were not available at the time of the study. In their work, which is a proof-of-concept study, Mazzoleni et al. (2017) showed that assimilation of CSD improves the overall model performance; the impact of irregular frequency of available CSD, and that of data uncertainty, were also deeply assessed. However, the use of synthetic CSD in conjunction with (semi-)distributed hydrological models deserves further discussion. As a result of equifinality, poor model identifiability, and lacks in model structure, internal states of (semi-)distributed models can hardly mimic the actual states of complex systems away from calibration points. Accordingly, the use of synthetic CSD that are drawn from model internal states under best-fit conditions can lead to overestimating the effectiveness of CSD assimilation in improving flood prediction. Operational flood forecasting, which results in decisions of high societal value, requires robust knowledge of the model behaviour and an in-depth assessment of both model structure and forcing data. Additional guidelines are given that are useful for the a priori evaluation of CSD for real-time flood forecasting and, hopefully, to plan apt design strategies for both model calibration and collection of CSD.

## 1 Introduction

Flood forecasting has a critical importance as it results in decisions of high societal value. In order to produce the most accurate flood predictions, it is essential to provide public authorities with the best combination of data and models, and with a robust knowledge of the model behaviour in terms of reliability and uncertainty. Modellers thus have a responsibility to deeply assess the strengths and limitations of model forcing data.

Within this general picture, the topic of community-based monitoring aimed at providing crowdsourced data (CSD) is gaining increasing attention among hydrologists (Le Coz et al., 2016; Walker et al., 2016; de Vos et al., 2017; Smith et al., 2017; Starkey et al., 2017). For example, the availability of hydrometric data, collected by active citizens in the course of severe flood events, offers a new, exciting chance to improve real-time flood forecasts. However, the use of CSD poses challenges to modellers since their information content, reliability, arrival frequency, and location are a priori unknown (Mazzoleni et al.,

2015, 2017; McCabe et al., 2017; van Meerveld et al., 2017; Yang and Kang, 2017). In addition, long time series of (CSD) are unavailable, thus complicating efforts to assess their effectiveness in improving flood prediction.

In pioneering applications (Mazzoleni et al., 2015), CSD collected in the upper part of a basin were assimilated into adaptive hydrological models to reduce uncertainty in forecasting flood hydrographs at downstream sections. In this recent work,

Mazzoleni et al. (2017) paid particular attention to the issues of uncertainty and irregular arrival frequency of CSD. Their results showed that assimilation of CSD improves the overall model performance. They also showed that the accuracy of CSD is, in general, more important than their arrival frequency.

In their work, the Authors used synthetic (i.e., not actually measured) CSD, because real streamflow CSD were not available at the time of the study. Commenting on this aspect, the Authors wrote "*the developed methodology is not tested with data*

*coming from actual social sensors. Therefore, the conclusions need to be confirmed using real crowdsourced observations of water level*". A practical verification of the results by Mazzoleni et al. (2017) is indeed necessary; furthermore, particular attention has to be paid to possible drawbacks inherent in the use of CSD for operational flood forecasting and related to model structural uncertainty, which are not discussed in their proof-of-concept study.

The Comment is outlined as follows. Section 2 presents an in-depth assessment of the Bacchiglione River case study (i.e.,

the fourth case study presented in Mazzoleni et al., 2017), in order to highlight the actual gap between a proof-of-concept study and a real application for operational flood forecasting. Given the complexity of the basin and the relatively paucity of available data, it is shown that the semi-distributed model used in Mazzoleni et al. (2017) is unable to properly represent the physics of the whole hydrological and hydraulic system, which affects the interpretation of the usefulness of CSD. Based on the key features delineated in Sect. 2, a more general assessment of CSD assimilation in (semi-)distributed hydrological models

is given in Sect. 3. A brief summary closes the Comment.

## 2   Specific comments

### 2.1   The Bacchiglione catchment closed at Ponte degli Angeli (Vicenza)

The catchment of the upper Bacchiglione River, closed at Ponte degli Angeli in the historical centre of Vicenza (Fig. 1), is located in the north of the Veneto Region, a plain that is fringed by the Alpine barrier at a distance of less than 100 km to the

north of the Adriatic Sea (Barbi et al., 2012).

With regard to the precipitation climatology, the southern part of this plain is the drier, with approximately 700–1000 mm of mean annual rainfall, whereas more than 2000 mm are measured close to the pre-alpine chain due to the interaction of the southerly warm and humid currents coming from the Mediterranean Sea with the mountain barrier (Smith, 1979). A significant portion of the annual rainfall often concentrates into very short periods of time in the form of what often turns out to be an

extreme event with deep convection playing a central role (Barbi et al., 2012; Rysman et al., 2016). As a consequence, severe flooding events have threatened agricultural and urban areas in the recent years (e.g. Viero et al., 2013; Scorzini and Frank, 2017).

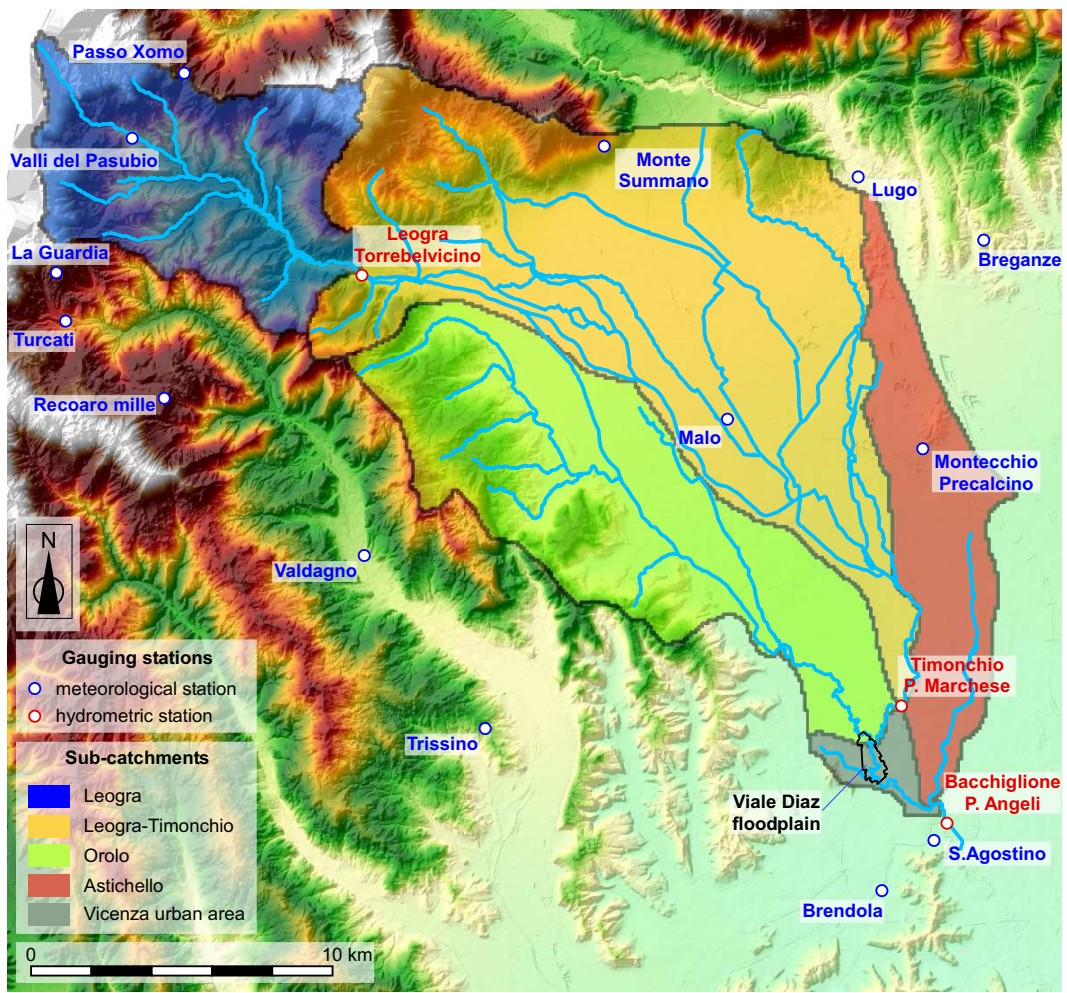

**Figure 1.** The catchment of the Bacchiglione River at Ponte degli Angeli, Vicenza (Italy).

Due to the spatial and temporal variability of the rainfall fields, meteorological models are often unable to provide accurate and reliable quantitative precipitation estimates (QPE) for the upper Bacchiglione catchment. An example of this inadequacy is given, for instance, by Fig. 13 in Mazzoleni et al. (2017).

The upper Veneto plain is a highly populated and urbanized area, with extremely complex drainage and irrigation networks that significantly affect both runoff production and propagation (Viero and Valipour, 2017). Within this plain, the Bacchiglione River and its tributaries are provided with relatively high levees (Viero et al., 2013), which prevent the exchange of water from inside to outside the riverbed (and vice versa) when the inner water levels are relatively high. As a consequence, the minor channel networks are not always allowed to deliver their drainage water towards the nearest tributary, i.e., the inflow points along the main river reaches change during a flood event depending on the instantaneous water level within the river. This

occurrence modifies the network connectedness which, in turn, leads to different mechanisms of hydrologic response in the overall catchment.

Just upstream of the City of Vicenza, an area of up to 1 km$^2$ (the "Viale Diaz" floodplain, Fig. 1) is flooded when the Bacchiglione flow rate exceeds $\sim$ 160 m$^3$/s. Since about $2\cdot10^6$ m$^3$ of water can be temporarily stored in this area, a significant flood attenuation can be produced, particularly in case of hydrographs with a steep rising limb (which is often the case due to the climatic regime and the catchment characteristics).

Moreover, the lower part of the Bacchiglione basin, North of Vicenza, includes a vast groundwater resurgence zone, in which it's difficult to assess both the actual contribution of resurgence to the Bacchiglione streamflow (up to $\sim$ 30 m$^3$/s) and the time-variable behaviour of soil moisture.

Clearly, such a system is highly non-linear. Nonetheless, significant parts of the Bacchiglione catchment are poorly monitored, and the remaining parts are completely unmonitored. The Leogra subcatchment (blue shaded area in Fig. 1) is provided with a pressure-transducer for the measure of water level at Torrebelvicino (Fig. 1). A rating curve derived from theoretical considerations is available for this cross-section. However, the absence of instrumental measures of flow discharge limits its reliability. The Leogra-Timonchio subcatchment (orange shaded area in Fig. 1) is monitored by an ultrasonic stage sensor located at Ponte Marchese, just upstream of the confluence with the Orolo River. Flow rate measurements at Ponte Marchese refers only to low hydraulic regimes, and show great variability due to the operation of a hydroelectric power plant located just downstream of Ponte Marchese. The Orolo River (green shaded area in Fig. 1), with a discharge capacity of more than one third of the Bacchiglione at Ponte degli Angeli, is one of its major tributaries. Unfortunately, not only the Orolo subcatchment is completely uncovered by meteorological gauging stations, but also no hydrometric gauging stations are present along its reach. Similarly to the Orolo, the Astichello catchment (red shaded area in Fig. 1) is unmonitored and, due to backwater effects, significant areas adjacent to the main channel of the Astichello are flooded when water levels in the Bacchiglione are relatively high. Hence, the discharge that effectively flows from the Astichello into the Bacchiglione River may significantly reduce depending on the water stage within the main course of the Bacchiglione River.

Attention must be paid to the fact that the three major tributaries (Orolo, Timonchio, and Astichello) meet just upstream of the gauging station of Ponte degli Angeli (Fig. 1), making it difficult to correctly estimate the actual contribution of each single tributary to the total streamflow. By looking at the tree-like structure of the drainage network in an electrical analogy (Rodríguez-Iturbe and Rinaldo, 2001), the major tributaries of the Bacchiglione are in fact "conductors in parallel".

Certainly, given the irregular topography of the catchments, the heterogeneity of the landscape, and the complexity of the hydraulic network, it can be stated that the Bacchiglione catchment is poorly monitored.

## 2.2   The semi-distributed model of the Bacchiglione catchment

In catchments like that of the Bacchiglione River, for all the reasons reported in the previous section, the accurate prediction of flood hydrographs with continuous time simulation is unquestionably a difficult task (Anquetin et al., 2010).

Mazzoleni et al. (2017) used an available semi-distributed hydrological model coupled with a Muskingum–Cunge scheme for flood propagation within the main river network, which was originally set up to forecast flood hydrographs of the Bac-

chiglione River at Ponte degli Angeli (Vicenza). Sensibly, the model was calibrated by minimizing the root mean square error between observed and simulated values of water discharge only at Ponte degli Angeli, which is the only hydrometric station provided with a reliable rating curve. The semi-distributed model, although explicitly representing the hydrological processes within the main subcatchments, has to be intended as a lumped model from a practical standpoint, since the discharge in Ponte degli Angeli is its only control point.

Therefore, no matter the accuracy of streamflow prediction in Ponte degli Angeli, little can be said about the accuracy of the model in describing the internal states of the system, such as the streamflow along upstream tributaries. This limitation has to be ascribed to uncertainty in precipitation fields, to the paucity of (reliable) flow rate data upstream of Vicenza, and to inherent limitations of the model itself.

Indeed, it has to be remarked that the Muskingum–Cunge model for flood propagation used in Mazzoleni et al. (2017) considers rectangular river cross-sections for the estimation of hydraulic radius, wave celerity, and other hydraulic variables (Todini, 2007). Accordingly, the effects exerted by the "Viale Diaz" floodplain, which acts as a sort of in-line natural flood control reservoir on flood propagation, can not be properly accounted for. This means that, if the flood hydrograph is correctly modelled at Ponte degli Angeli, it can not be correctly modelled upstream of the Viale Diaz floodplain (and vice versa).

## 2.3   The use of synthetic CSD in the Bacchiglione case study

In the Bacchiglione case study, Mazzoleni et al. (2017) calibrated the model using measured rainfall data to well reproduce the streamflow hydrograph at the basin outlet (call this post-event simulation "scenario 1"). Then they forced the model with predicted rainfall fields that were completely different from the actual storm event ("scenario 2"); in this case, the discharge simulated using forecasted input was very different from that obtained using recorded rainfall, with a significant time shift and errors in predicted discharge ranging between 25 and 50% at the flood peak (and up to 90% if considering synchronous data). In their "scenario 3", similarly to the "observing system simulation experiment" (OSSE) approach, synthetic streamflow CSD extracted from the "scenario 1" were assimilated into a new run using the same forcing as in the "scenario 2". Not surprisingly, the model performance in the "scenario 3" was significantly better than in the "scenario 2", as the synthetic CSD they assimilated were representative of the model internal states in the best-fit scenario.

The Authors argued that the synthetic CSD they used are realistic. For this condition to be met, given that these CSD are the results of the model itself, the model must well represent the physics of the real system (i.e., it must be calibrated or, at least, verified) at locations where CSD are first generated and then assimilated; this is a fundamental hypothesis behind the OSSE approach. The synthetic CSD used in Mazzoleni et al. (2017) for the Bacchiglione case study are drawn from the model internal states under best-fit conditions. Thus, when the model is forced with different (wrong) input data, their assimilation is expected to be as successful as possible in updating the model states toward the best-fit scenario. However, the accuracy of such synthetic CSD is questionable, since they do not refer to model control points (i.e., they are drawn from the semi-distributed model at locations where the model is neither calibrated nor verified), so nothing can actually be said about the model performance at these locations. In a sense, synthetic CSD used by Mazzoleni et al. (2017) are *optimal* (in view of assimilation performance)

rather than *realistic*. Since real CSD are likely biased with respect to the synthetic CSD actually used, assimilation of real CSD can not be as effective as that performed in Mazzoleni et al. (2017).

From one point of view, it is possible that such an inconsistency could have led Mazzoleni et al. (2017) to overrate the importance of CSD, as they considered issues related to CSD precision, but not accuracy (Mazzoleni et al., 2016). Therefore,
additional care must be taken in operational flood forecasting when assimilating CSD into (semi-)distributed hydrological models at locations other than model control points.

## 3  The use of real CSD in operational flood forecasting

As remarked by Mazzoleni et al. (2017), the success of assimilating real CSD in hydrological modelling strictly depends on their accuracy, quantity, and spatial-temporal distribution. However, this comment points out that attention must be paid not
only to CSD, but also to the model.

In general, historical data recorded by traditional sensors are first used to calibrate a model; then, in real-time mode, the same sensors provide data both to force the model and to update the model states (e.g., Ercolani and Castelli, 2017); moreover, the reliability of data from traditional sensors outperforms that of CSD. Hence, from a practical point of view, CSD have limited usefulness at locations already equipped with traditional sensors. Since their natural purpose is to enhance (rather than replace)
data from traditional sensors, and considering that they can be collected at locations not known a priori, CSD typically do not refer to model calibration points.

Given the spatially distributed nature of CSD, spatially explicit hydrological models can take the major advantage from CSD. On the other hand, particular care has to be taken when dealing with physically based, (semi-)distributed models, which are known to suffer from equifinality and poor identifiability of model parameters (Beven, 2006).
After the critical work by Beven (1989), detailed investigations were carried out about the model complexity needed to simulate rainfall-runoff process. Several studies indicated that the information content in a rainfall-runoff record is sufficient to support models of only very limited complexity (Jakeman and Hornberger, 1993; Refsgaard, 1997). This implies that distributed, or semi-distributed, hydrological models are seldom calibrated; rather, they are commonly over-parametrized, since calibration rarely involves their internal states (Sebben et al., 2012; Viero et al., 2014).
In addition, flood routing processes are typically oversimplified in operational models meant to real-time flood forecasting (Mejia and Reed, 2011). For instance, significant effects related to either compound sections, large floodplains connected to the main channel, or confluences causing backwater effects, are seldom accounted for.

As a consequence, (semi-)distributed rainfall-runoff models may provide accurate predictions of outflow discharge at the basin outlet and, at the same time, poor predictions of internal states of the system (e.g., the soil moisture content, or the relative
contribution of upstream tributaries); in other words, one can likely get the correct answer for the wrong reason (Loague et al., 2010). Therefore, (semi-)distributed models can be said calibrated only at calibration (or control) points, and verified only at locations in which model results are shown to compare favourably with enough (and accurate enough) measured data.

This caveat particularly applies to assimilation of CSD in hydrological modelling for operational, real-time flood forecasting. Indeed, while CSD typically refer to model internal states, they are assimilated in order to improve the accuracy of the main outputs of the model, such as streamflow hydrographs at basin outlet (model internal states are relatively less important in this context).

Recalling that model input, states, parameters, and outputs (or a subset of them) can be updated using different data assimilation techniques (Refsgaard, 1997), assimilation of CSD in operational flood forecasting can be helpful provided that the model is able to well represent the physics of the system at locations where CSD are collected. Of course, data assimilation can contribute, in many cases, to improve such a representation. However, when only internal states are updated (as in Mazzoleni et al., 2017), this condition is met if (and only if) the model is properly calibrated and verified at locations where
CSD refer to. Otherwise, correcting internal states of a poorly calibrated model can even lead, in principle, to worse predictions at the outlet than performing no corrections at all (Crow and Van Loon, 2006). It is undoubtedly difficult to assess this issue when only synthetic CSD, generated by the same model, are available for testing the overall method.

As an alternative for operational forecasting, ensemble based data assimilation methods (e.g., the Ensemble Kalman Filter or the Particle Filter) can be used to update jointly model states and parameters and to provide a direct measure of uncertainty
(Moradkhani et al., 2005; Salamon and Feyen, 2009; Wani et al., 2017). In this way, models cope directly with equifinality and problems of over-parametrization, since parameter posterior distributions are represented by ensembles. Note that typical data assimilation algorithms are in principle able to screen out noisy data automatically, but need to be modified to tackle possible data bias, which otherwise leads to poorly calibrated models. Thus, it is important, regardless of the nature of the data, to verify if such bias exists before any data assimilation is applied.

Nonetheless, also such sophisticated tools may fail if the model has structural deficiencies that make it unable to represent true system states at given locations. As a representative example, consider the Bacchiglione River (Fig. 1) and, specifically, the "Viale Diaz" floodplain described in Sec. 2. The role played by such an in-line flood control reservoir on flood routing can not be accounted for using a basic Muskingum–Cunge model that considers rectangular cross-sections. It follows that the assimilation of accurate streamflow data referring to a section located just upstream of the Viale Diaz floodplain (e.g., Ponte
Marchese, see Fig. 1) can likely deteriorate the model predictions in Ponte degli Angeli, downstream of the floodplain.

Shortcomings similar to the one described above, which can be found in many different case studies, can be a priori conjectured through a close inspection of both the physical system and the model characteristic. Their quantitative assessment needs an extensive comparison with measured data; of course, a "blind" use of CSD (i.e., their assimilation at locations where the model is neither calibrated nor verified) is at least questionable.

**4   Summary**

The approach proposed and investigated by Mazzoleni et al. (2017), based on the assimilation of crowdsourced data (CSD), can be generally valuable to improve real-time flood forecasts using non-traditional information now available thanks to active citizens and new technologies.

However, it has to be remarked that physically based modelling of rainfall-runoff and flow routing processes face limitations ascribed to the paucity of measured data, to the complexity of real environments, and to lacks in model structure and parametrization. As a consequence, (semi-)distributed rainfall-runoff models used for operational flood forecasting can provide reliable predictions at locations where calibration is performed (i.e., control points) and, at the same time, incorrectly represent
5 system states elsewhere (e.g., discharges in upstream, ungauged tributaries).

In a context of equifinality and simplified representation of real physical processes, the accurate prediction of outflow hydrographs can be achieved even though model internal states don't match the true system states. In such cases, the assimilation of real CSD can lead to a substantially lower performance than the use of synthetic CSD would suggest, as it corresponds, in fact, to update a model using biased data (e.g., Dee, 2005; Liu et al., 2012). When only internal states (and not model parameters)
10 are updated, or when the model suffers structural deficiencies, the assimilation of real (i.e., not synthetic) streamflow data at internal points can lead, in principle, to even worse model prediction at the outlet than no assimilation at all (Crow and Van Loon, 2006). The problem can arise due to the disjoint use of traditional and crowdsourced data, with the former used to calibrate (semi-)distributed models at control points, and the latter used only in real-time to update model states at different locations.

A possible solution is the use of ensemble based data assimilation methods to update jointly model states and parameters.
15 An additional pragmatic recommendation is the collection of accurate measured data for a suitable period, for at least two reasons: i) to develop reliable rating curves at locations where water level CSD are planned to be collected, and ii) to calibrate and verify the model ability in describing the system states correctly at the locations in which CSD are collected.

It must be observed that, while scarce control on the collection of CSD can be exerted during significant flood events, the locations at which citizens can collect CSD of water levels is always determined a priori, since the availability of rating curves
20 is a necessary condition in order to convert water levels into discharges. The amount of measured data needed to develop reliable rating curves can also be profitably used to calibrate the model at those sections as well.

As a final remark, both modellers and environmental agencies should comprehensively account for the characteristics of the physical system, for model structure and parametrization, for the design of the sensor network, and for data to be used both in calibration and in operational mode.

25 *Acknowledgements.* The Editor, M. Mazzoleni and two anonymous reviewers are gratefully acknowledged for providing valuable comments and suggestions that allowed to significantly improve the manuscript.

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
