# Peer review of "Comment on "Can assimilation of crowdsourced data in hydrological modelling improve flood prediction?" by Mazzoleni et al. (2017)"

_Hydrology and Earth System Sciences, 2017_

## Referee Comment (RC2) · Anonymous Referee #2 · 25 Apr 2017

The author makes some significant critical remarks on the work of Mazzoleni et al. (2017) that are worth to be considered for publication. However, I would first advise to mellow the tone of the narrative. In addition, I invite the author to make sure that the comments are more general and less focused on the upper Bacchiglione river catchment presented by Mazzoleni et al. (2017). In doing so, Section 2.1 should be reduced considerably, as most of the information and comments seem too specific, and might not be supported for the other test sites. The paper of Mazzoleni et al. (2017) aimed at investigating the value of information retained by crowdsourced data (CSD) when assimilated in surface flow models for flood prediction. Their work is admittedly a proof-of-concept study and the synthetic feature of CSD is quite clear, rather than

"briefly mentioned". Their conclusions are correct so long as one assumes the model well represents the physics of the hydrological system, which is a fundamental hypothesis behind observation simulation system experiments. On the other hand, I agree that there seems to be an inherent tendency in Mazzoleni et al. (2017) to present results in a way that somehow overstates the importance of CSD. There are, in my view, some major points that need to be highlighted: • The method chosen for calibrating a model should be consistent regardless of the type of data used. For non-linear models, ensemble based data assimilation methods (e.g the EnKF or the PF) are attractive in that they can be used to update jointly model states and parameters and provide a direct measure of uncertainty. Note that these models cope directly with problems of over parameterization and equifinality since parameter posterior distributions are represented by ensembles. • CSD can be instrumental to reduce model uncertainty. Indeed, one can assimilate these data together with traditional hydrologic observations, thereby reducing parameter uncertainty even in those regions where the original reliability of the model is inadequate. In general, the value of information of these data is strictly dependent on their quantity, quality, spatial-temporal distribution. Note that typical data assimilation algorithms are in principle able to screen out noisy data automatically, but need to be modified to tackle possible data bias, which otherwise leads to poorly calibrated models. Thus, it is important, regardless of the nature of the data, to verify if such bias exists before any data assimilation is applied.

---

## Author Comment (AC1) · 20 May 2017

I'm grateful to the reviewers for their suggestions, which are actually helpful. I'm confident that the revision of the paper will lead to a more focused comment, allowing to better order its structure, and possibly enriching the paper in terms of contents.

**1   RESPONSE TO THE COMMENTS OF REVIEWER #1**

*The comment on "Can assimilation of crowdsourced data in hydrological modelling improve flood prediction?" addresses the subtle drawback hidden behind the practice*

*of using traditional and crowdsourced data, recorded at different locations, disjointly. The former are used to calibrate semi-distributed models and to force them in real-time, the latter only to update the model states in operational forecasting.*

*In Mazzoleni et al. (2017), synthetic CSD were generated as model results using observed precipitation, while simulated results were obtained using forecasted precipitation. Since the semi-distributed hydrological model used in* ? *was calibrated at only one location, Viero (2017) underlined that synthetic CSD at interior points (different from the calibration one) cannot be considered reliable due to equifinality issues. In fact, semi-distributed hydrologic models are commonly over-parametrized and may provide accurate predictions only where the model is calibrated, and it can fail to represent the relative contribution of upstream tributaries. I read the comment with interest and I really appreciate all the author's efforts. However, I have many doubts and considerations that I would like to share with him.*

*Maurizio Mazzoleni*

I thank Maurizio Mazzoleni (Reviewer #1) for his appreciation of the Comment and for his valuable comments and suggestions, which are addressed in the following.

1. *Overall, I found that the main message of the author have been stretched and repeated many times throughout the Comment.*

   I agree. This is due, in part, to the brevity of the Comment; the main message is repeated at least in the Abstract, in the Introduction, and in the Summary. In the revised version of the Comment, I'll try to better organize the text in order to avoid (or, at least, to limit) useless repetitions.

2. *It is not clear to me what would the author propose to generate synthetic CSD when only measurements from traditional sensors, located at points different from*

*the ones of CSD, are available. In the summary section, only a pragmatic solution is suggested in case of availability of distributed flow data (not the case in Mazzoleni et al., 2017). This solution involves the collection of CSD for a suitable test period, to verify the model ability in describing the system states correctly at the locations in which CSD are collected. However, this solution will open many other types of questions. For example, how would the author assess the quality of the CSD? Which category of citizen the author would engage in order to collect CSD? For how long will this data collection take place? How can it be insured that CSD quality during data collection will be the same as the CSD quality during real-time modelling updating (no control)? Citizens accuracy is different and data quality assessment is still a burning topic in citizen sciences. In addition, CSD in calibration may be different from the ones in real-time model updating.*

The work by Mazzoleni et al (2017) is actually a proof-of-concept, which analyze major aspects of the assimilation of crowdsource data in order to improve the forecasting of hydrological models. My Comment essentially focuses on what should follow a proof-of-concept, i.e., on the use of actual crowdsourced data in real, operational flood forecasting. Indeed, it is the passage from a proof-of-concept study to a real-world application (i.e., from synthetic CSD to actually measured CSD) that entails the additional significant drawbacks related to equifinality, overparameterization, and deficiency in model structure, which are not discussed in Mazzoleni et al. (2017).

Accordingly, my Comment is not specifically aimed at proposing a different, better way to generate synthetic CSD when measurements from traditional sensors are available only at different locations from the ones of CSD, as this would mainly pertain to the proof-of-concept study.

I agree that the solution proposed in my Comment opens many other questions, but a problem do exists with when assimilating CSD referring to locations in which the model states can substantially differ from real states. The existence of this

problem cannot be ignored; rather, being aware of it is per se important (this is one of the most important reason behind my comment).

Thanks to the suggestion by Reviewer #2, in the revised version of the Comment I'm going to suggest an additional possible solution, which can potentially solve (or aid solving) other questions raised by M. Mazzoleni (e.g., quality of CSD that can be different in calibration and in operational use, etc.). This solution is the use of ensemble-based data assimilation methods to update jointly model states and parameters (and not only model states). Again, like the rest of the Comment, this solution refer to operational use of hydrological models with real (i.e., not synthetic) CSD.

3. *Moreover, I do not understand to which extent the comments of the Author are referred to the paper of Mazzoleni et al. (2017) or to a generic issue on the use of CSD in hydrological modelling.*

I'm aware that it is actually difficult to properly balance comments that must be specific (in that they refer to particular aspects of a given work) and, at the same time, they should be significant in a wider sense. Consider that Reviewer #2 criticized the Comment (in particular Section 2.1) as too specific. I'll try to find a better equilibrium between specificity and generality in the revised version of the Comment.

4. *The Author mentioned that "Indeed, for synthetic streamflow CSD to be realistic, two specific requirements have to be met: i) a reliable rating curve must be available for the cross sections where hydrometric CSD are recorded, and ii) the model has to be calibrated at these locations". I agree with the author in case of CSD provided by static sensors, like in case of Mazzoleni et al. (2017). However, in a real scenario where CSD are provided by citizens at random moments and locations within the catchment, by means of dynamic sensors, I do not agree with the second point of the comment for two reasons. Firstly, assuming the author*

*is right, it would be extremely difficult to calibrate the model with observed data
at unknown locations in which synthetic CSD will be assimilated. Secondly, it is
not clear to me why synthetic CSD based on model results should be generated
if observed data are already available at the CSD/calibration points. Obviously,
such observed data should be directly used to generate synthetic scenarios of
CSD, like in case of the first three case studies in Mazzoleni et al. (2017), without
using any model result.*

I thank M. Mazzoleni for this comment, which help me to clarify the focus and
the structure of my Comment. In the revised version of the Comment, I'm going
to better separate comments referring to the reliability of synthetic CSD due to
equifinality issues, from comments on the use of actual CSD in operational fore-
casting. In this way, I'm confident that misunderstandings, such that those here
underlined, could be removed.

5. *Another extremely important point is the assimilation of CSD observations. From
Viero's Comment, it is not clear how erroneous synthetic observations can affect
assimilation performances. The author briefly mentions this issue referring to
Dee (2005) and Liu et al. (2012). Honestly, since the main objective of Mazzoleni
et al. (2017) was the assimilation of realistic synthetic CSD, I was expecting
a more comprehensive analysis on the effect of assimilating biased/uncertain
observations within hydrological model.*

The issue raised by M. Mazzoleni is undoubtedly interesting; although being not
the primary objective of my Comment, it deserve further discussion. Neverthe-
less, it seems to me that this specific criticism descends from the fact that one of
the main goals of Mazzoleni et al. (2017) was how to generate realistic synthetic
CSD (this issue pertains to the proof-of-concept scope), whereas I am mainly
interested in the differences that modelers may find when moving from proofs-
of-concept to real-world applications. I remain convinced that it is impossible to
determine if synthetic CSD are realistic or not, when these CDS are generated

using a hydrological model at locations where it is impossible to calibrate/verify
this model.

In the revised version of the Comments, I'll try to better explain this issue, which is
actually intricate. Measured CSD are obviously affected by uncertainty (this issue
is assessed in Mazzoleni et al., 2017), and by bias defined as systematic devia-
tion from the actual values (this issue is assessed, e.g., in Dee, 2005 and in Liu
et al., 2012, but also in Crow and Van Loon, 2006, who stated that *"inappropriate
model error assumptions can lead to circumstances in which the assimilation of
surface soil moisture observations actually degrades the performance of a land
surface model"*).

In my Comment, I want to point out that even accurate and unbiased measured
data can be "seen" as biased data by a model. This can occur when the model is
not properly calibrated at sections where data refer to (and model parameters are
not update along with model states), or when the model is unable to reproduce
the actual dynamic of the system at that location due to intrinsic limitations of
the model structure. This issue is better explained with practical examples in my
answer to the following point #6.

6. *In addition, Viero stated, "In a context of equifinality and of poor identifiability of
model parameters, the model internal states can hardly mimic the actual sys-
tem states away from calibration points, thus reducing the chances of success in
assimilating real (i.e. not synthetic) CSD." Why the chances of success in assim-
ilating real CSD is reduced if the model is not calibrated at CSD location? Does
this mean that in case of assimilation of distributed soil moisture observations
from remote sensing, within a distributed hydrological model, we would need to
calibrate the model in each grid cell? I disagree with the author. The main pur-
pose of data assimilation is to use real-time (noisy) observations to update the
wrong estimate of the states of a dynamic model (not able to mimic the actual
system states away from calibration point). Assimilation of observations at inter-*

*nal points of the catchment is very useful when model states are less accurate than real-time observations.* **If a model is able to correctly predict actual system states away from calibration points, why should someone bother to add complexity and uncertainty assimilating CSD observations?** *The literature provides many studies (e.g. Rakovec et al., 2012) in which hydrological models are updated using measurements at internal points, even if such observations are not used during model calibration.*

Thank you for this comment. I realize that I was not precise enough in that part of my Comment, which should be improved.

To answer the key question in this Reviewer's comment (which I highlighted above), I remark that a model can predict wrong system states away from calibration points for different reasons (e.g., wrong/insufficient input data and/or poor calibration and/or structural model deficiencies). Assimilation of observations at internal points of the catchment can be extremely useful when model states are less accurate than real-time observations, but not when this lack of accuracy of model states is due to problems with model structure (or due to poor calibration of model parameters if such parameters are not updated through data assimilation along with the model states).

Therefore, I stress that the statements by M. Mazzoleni are reasonable, but they implicitly assume that the model structure (and the set of model parameter as well, since they were not updated through data assimilation in his work) is (are) able to correctly estimate both the internal states and the model outputs. Although this is desirable for physical-based models (see also the comment #3 of Reviewer #2), one must admit that it is not true in general and, reasonably, it is not true for the model application of the Bacchiglione River presented in Mazzoleni et al. (2017).

I try to clarify the question using first a hypothetical example. Consider a hydrological model, not calibrated at internal points, which provides the right prediction at the closing section as the result of wrong predictions at some internal points. The updating of model states at this internal points based on real data (i.e., different to the internal states needed to provide the 'correct' prediction at the outlet) will likely cause this model to produce worse predictions at the closing section with respect to no assimilation at all. This possible occurrence cannot be detected, nor assessed, if data to be assimilated are extracted from the model itself, because in this case the synthetic data represents the wrong internal states (with respect to the real data at these points).

The problem of assimilating data not coherent with internal model states (when this is due to poor estimation/identifiability of model parameters) could be limited by updating the model parameters along with the internal states of the model (as suggested by Reviewer #2), but this strategy could not be sufficient if the model has structural deficiencies.

As a practical example, consider the "Viale Diaz" floodplain, described in my Comment, which acts as a sort of in-line natural flood control reservoir on flood propagation. Since the attenuation of flood wave exerted by this floodplain can not be properly accounted for by the routing model used in Mazzoleni et al. (2017), the (hypothetical) assimilation of a correct flood hydrograph upstream of the Viale Diaz floodplain leads to incorrect predictions at Ponte degli Angeli, downstream of the "Viale Diaz" floodplain.

7. *I am puzzled with the sentence "Therefore, beside the key points identified by Mazzoleni et al. (2017), not only data, but also the model has to match specific requirements for data assimilation to be successful". What are these specific requirements that model has to match? Is the Author referring to the reliability of synthetic data at calibration points and to the capability of the model to represent truth states?*

The need to assimilate suitable crowdsourced data was assessed in Mazzoleni et al. (2017). With respect to the specific requirements that the model has to

match, its ability of well representing the physics of the hydrological system (i.e., of correctly representing true internal states when forced by correct input data) is actually the key aspect. I'll try to make this point clearer in the revised version of the Comment.

**2   RESPONSE TO THE COMMENTS OF REVIEWER #2**

*The author makes some significant critical remarks on the work of Mazzoleni et al. (2017) that are worth to be considered for publication.*

I thank Reviewer #2 for his/her appreciation of my Comment and for his/her very useful and precise suggestions, which are addressed in the following

1. *However, I would first advise to mellow the tone of the narrative.*

   Thanks for the suggestion. I'll revise the text of the paper, trying to smooth the English (and to fix typos).

2. *In addition, I invite the author to make sure that the comments are more general and less focused on the upper Bacchiglione river catchment presented by Mazzoleni et al. (2017). In doing so, Section 2.1 should be reduced considerably, as most of the information and comments seem too specific, and might not be supported for the other test sites.*

   I thank the reviewer for his suggestion. I'll try to shorten Section 2.1 in the revised version of the Comment. While I agree that Sect. 2.1 is very specific, I do believe that most part of this specificity is not meaningless for other test sites. Indeed, I remain convinced that much can be learned from in-depth analyses of specific cases. The opposite risk is the (often unperceived) oversimplification of

   real systems and processes in our schematic representations (i.e., models) of the reality.

   Besides its evident specificity, one of the goals of Sect. 2.1 is to highlight that real-world case studies are often far more complex than what can emerge from most of the applications reported in the literature (this is undoubtedly due to actual limits in papers' length). I am convinced that hydrologists can easily find similarities with other case studies.

   Finally, the Comment is indeed a comment to a specific paper, and only one of the four model applications reported by Mazzoleni et al (2017) is here commented, since the contents of the Comment only apply to semi-distributed (and over-parameterized) models and to the assimilation of CSD data in location where the model cannot be calibrated. In the other test cases presented in Mazzoleni et al (2017), the Authors used a lumped model and assimilated CSD only at the calibration sections.

3. *The paper of Mazzoleni et al. (2017) aimed at investigating the value of information retained by crowdsourced data (CSD) when assimilated in surface flow models for flood prediction. Their work is admittedly a proof-of-concept study and the synthetic feature of CSD is quite clear, rather than "briefly mentioned". Their conclusions are correct so long as one assumes the model well represents the physics of the hydrological system, which is a fundamental hypothesis behind observation simulation system experiments. On the other hand, I agree that there seems to be an inherent tendency in Mazzoleni et al. (2017) to present results in a way that somehow overstates the importance of CSD.*

   I agree with the reviewer. The fact that a model well represents the physics of the hydrological system is a fundamental hypothesis for physically-based models, and is tacitly assumed in Mazzoleni et al. (2017). However, it must be stressed that this requirement is not necessarily matched when semi-distributed, physically based models are actually used as lumped models, i.e., they are calibrated

only at the closing sections. Given the complexity of the Bacchiglione catchment, the relatively paucity of measured data, and the structure of the model used (see my answer to comment #6 of Reviewer #1 for further details), reasonably it is not true for the model application of the Bacchiglione River presented in Mazzoleni et al. (2017).

4. *There are, in my view, some major points that need to be highlighted: the method chosen for calibrating a model should be consistent regardless of the type of data used. For non-linear models, ensemble based data assimilation methods (e.g the EnKF or the PF) are attractive in that they can be used to update jointly model states and parameters and provide a direct measure of uncertainty. Note that these models cope directly with problems of over parameterization and equifinality since parameter posterior distributions are represented by ensembles. CSD can be instrumental to reduce model uncertainty. Indeed, one can assimilate these data together with traditional hydrologic observations, thereby reducing parameter uncertainty even in those regions where the original reliability of the model is inadequate. In general, the value of information of these data is strictly dependent on their quantity, quality, spatial-temporal distribution. Note that typical data assimilation algorithms are in principle able to screen out noisy data automatically, but need to be modified to tackle possible data bias, which otherwise leads to poorly calibrated models. Thus, it is important, regardless of the nature of the data, to verify if such bias exists before any data assimilation is applied.*

I thank the Reviewer for these interesting considerations. Ensemble based data assimilation methods are indeed powerful tools. On one hand, their use to jointly update model states and parameters can effectively circumvent the problem of uncertainty in model internal states at crowdsourced data points; on the other hand, such methods can help diagnosing poor identifiability of model parameters.

However, sophisticated tools to update jointly model parameters and states may fail if assimilating data in locations where the model is unable to correctly reproduce the actual physics of the system. While this possible occurrence can be a-priori conjectured through a close inspection of both the physical system and the model characteristic/capabilities, it can be proved (and quantified) only by comparing model results with measured data (i.e., model validation). The "blind" use of CSD (i.e., its assimilation at locations where the model is neither calibrated nor verified) is at least questionable (see, e.g., the examples reported in the answer to comment #6 of Reviewer #1).

Finally, in the Reviewer's comment it is stressed the importance of detecting bias in data to be assimilated. This observation pertains also to the object of my Comment, since real (i.e., not synthetic) data referring to locations where the model is unable to reproduce the physic of the system are equivalent to biased data for an model providing accurate estimates at these sections.

I'm going to add this considerations in the revised version of the Comment.

**References**

Crow, W. T. and Van Loon, E.: Impact of Incorrect Model Error Assumptions on the Sequential Assimilation of Remotely Sensed Surface Soil Moisture, J. Hydrometeorol., 7, 421–432, doi:10.1175/JHM499.1, 2006.

Dee, D.: Bias and data assimilation., Q. J. R. Meteorol. Soc., 131, 3323-3343, doi:10.1256/qj.05.137, 2005.

Liu, Y., Weerts, A., Clark, M., Hendricks Franssen, H.-J., Kumar, S., Moradkhani, H., Seo, D.-J., Schwanenberg, D., Smith, P., van Dijk, A., van Velzen, N., He, M., Lee, H., Noh, S., Rakovec, O., and Restrepo, P.: Advancing data assimilation in operational hydrologic forecasting: progresses, challenges, and emerging opportunities, Hydrol. Earth Syst. Sc., 16, 3863–3887, doi:10.5194/hess-16-3863-2012, 2012.

Mazzoleni, M., Verlaan, M., Alfonso, L., Monego, M., Norbiato, D., Ferri, M., and Solomatine, D.: Can assimilation of crowdsourced data in hydrological modelling improve flood prediction?, Hydrol. Earth Syst. Sc., 21, 839–861, doi:10.5194/hess-21-839-2017, 2017

Viero, D.P: Comment on "Can assimilation of crowdsourced data in hydrological modelling improve flood prediction?" by Mazzoleni et al. (2017), Hydrol. Earth Syst. Sci. Discuss., doi:10.5194/hess-2017-102, in review, 2017.

---

## Author Response (AR1)

**1 Point-by-point reply to comments**

I'm grateful to the Editor and the Reviewers for their suggestions, which I found insightful and helpful. I'm confident that the revised version of the paper, with respect to the previous version, is more clearly focused, ordered in its structure, and enriched in terms of contents.

According to the Editor's suggestion, I paid particular attention to clarifying that the comment is about the operational aspect of using crowdsourced data.

**2 RESPONSE TO THE COMMENTS OF REVIEWER #1**

*The comment on "Can assimilation of crowdsourced data in hydrological modelling improve flood prediction?" addresses the subtle drawback hidden behind the practice of using traditional and crowdsourced data, recorded at different locations, disjointly. The former are used to calibrate semi-distributed models and to force them in real-time, the latter only to update the model states in operational forecasting.*

*InMazzoleni et al. (2017), synthetic CSD were generated as model results using observed precipitation, while simulated results were obtained using forecasted precipitation. Since the semi-distributed hydrological model used in Mazzoleni et al. (2017) was calibrated at only one location, Viero (2017) underlined that synthetic CSD at interior points (different from the calibration one) cannot be considered reliable due to equifinality issues. In fact, semi-distributed hydrologic models are commonly over-parametrized and may provide accurate predictions only where the model is calibrated, and it can fail to represent the relative contribution of upstream tributaries. I read the comment with interest and I really appreciate all the author's efforts. However, I have many doubts and considerations that I would like to share with him.*

*Maurizio Mazzoleni*

I thank Maurizio Mazzoleni (Reviewer #1) for his appreciation of the Comment and for his valuable comments and suggestions, which are addressed in the following.

1. *Overall, I found that the main message of the author have been stretched and repeated many times throughout the Comment.*

   I agree. In the revised version of the Comment, the text is substantially reorganized to limit useless repetitions.

2. *It is not clear to me what would the author propose to generate synthetic CSD when only measurements from traditional sensors, located at points different from the ones of CSD, are available. In the summary section, only a pragmatic solution is suggested in case of availability of distributed flow data (not the case in Mazzoleni et al., 2017). This solution involves the collection of CSD for a suitable test period, to verify the model ability in describing the system states correctly at the locations in which CSD are collected. However, this solution will open many other types of questions. For example, how would the author assess the quality of the CSD? Which category of citizen the author would engage in order to collect CSD? For how long will this data collection take place? How can it be insured that CSD quality during data collection will be the same as the CSD quality during real-time modelling updating (no control)? Citizens accuracy is different and data quality assessment is still a burning topic in citizen sciences. In addition, CSD in calibration may be different from the ones in real-time model updating.*

   The work by Mazzoleni et al (2017) is actually a proof-of-concept, which analyses major aspects of the assimilation of crowdsourced data in order to improve flood forecasting. My Comment essentially focuses on what should follow a proof-of-concept, i.e., on the operational use of CSD. Indeed, it is the passage from a proof-of-concept study to a real-world application (i.e., from synthetic CSD to actually measured CSD) that entails additional, significant drawbacks related to equifinality, overparametrization, and deficiency in model structure, which are not discussed in Mazzoleni et al. (2017).

Accordingly, my Comment is not aimed at proposing a different, better way to generate synthetic CSD when measurements from traditional sensors are available only at different locations from the ones of CSD, as this would mainly pertain to a proof-of-concept study.

In the revised version of the Comment, it should be clearer that its main aim is to highlight additional drawbacks inherent the use of CSD in operational flood forecasting, not assessed in Mazzoleni et al. (2017). Possible solutions and additional guidelines are now enhanced (thanks also to suggestions by Reviewer #2) and better explained.

3. *Moreover, I do not understand to which extent the comments of the Author are referred to the paper of Mazzoleni et al. (2017) or to a generic issue on the use of CSD in hydrological modelling.*

I am aware that it is actually difficult to properly balance comments that must be specific (in that they refer to particular aspects of a given work) and, at the same time, they should be significant in a wider sense. I revised the paper in order to make a better equilibrium by reporting specific comments in Section 2, and by debating them from a more general point of view in Section 3.

4. *The Author mentioned that "Indeed, for synthetic streamflow CSD to be realistic, two specific requirements have to be met: i) a reliable rating curve must be available for the cross sections where hydrometric CSD are recorded, and ii) the model has to be calibrated at these locations". I agree with the author in case of CSD provided by static sensors, like in case of Mazzoleni et al. (2017). However, in a real scenario where CSD are provided by citizens at random moments and locations within the catchment, by means of dynamic sensors, I do not agree with the second point of the comment for two reasons. Firstly, assuming the author is right, it would be extremely difficult to calibrate the model with observed data at unknown locations in which synthetic CSD will be assimilated. Secondly, it is not clear to me why synthetic CSD based on model results should be generated if observed data are already available at the CSD/calibration points. Obviously, such observed data should be directly used to generate synthetic scenarios of CSD, like in case of the first three case studies in Mazzoleni et al. (2017), without using any model result.*

I thank M. Mazzoleni for this comment, which help me to clarify the focus and the structure of my Comment. The revised version of the Comment more specifically deals with aspects related to operational use of CSD. In the Summary of the Comment, I remarked that locations at which streamflow CSD can be collected are actually always a-priori known because of the need of a rating curve, which must be developed before the assimilation of CSD in real-time operational use. Data collected in order to develop the rating curve should also be used to calibrate or, at least, to verify the model at these sections. Without actual rating curves, assimilation of synthetic CSD remains a theoretical exercise. Accordingly, while synthetic CSD can be certainly useful to carry out proof-of-concept studies and preliminary investigations, operational flood forecasting needs to rely on real data.

5. *Another extremely important point is the assimilation of CSD observations. From Viero's Comment, it is not clear how erroneous synthetic observations can affect assimilation performances. The author briefly mentions this issue referring to Dee (2005) and Liu et al. (2012). Honestly, since the main objective of Mazzoleni et al. (2017) was the assimilation of realistic synthetic CSD, I was expecting a more comprehensive analysis on the effect of assimilating biased/uncertain observations within hydrological model.*

The issue raised by M. Mazzoleni is undoubtedly interesting and deserves further discussion. The main objective of Mazzoleni et al. (2017) was the assimilation of realistic synthetic CSD. In my Comment, I explain that little can actually be said on the reliability of the synthetic CSD of the Bacchiglione case study. In the revised version, this aspect is better explained through a practical example related to the effects of the "Viale Diaz" floodplain on flood routing. The reader can grasp from this example which are the possible effects of assimilating biased data at location where the model is not verified and when only model states are updated (or in the case of structural deficiencies of the model).

Importantly, in my Comment I point out that even accurate and unbiased measured data can be "seen" as biased data by a model. This can occur when the model is not properly calibrated at sections where data refer to (and model parameters are not update along with model states), or when the model is unable to reproduce the actual dynamics of the system at those locations due to intrinsic limitations of the model structure. This issue is better explained with practical examples in my answer to the following point #6.

6. *In addition, Viero stated, "In a context of equifinality and of poor identifiability of model parameters, the model internal states can hardly mimic the actual system states away from calibration points, thus reducing the chances of success in assimilating real (i.e. not synthetic) CSD." Why the chances of success in assimilating real CSD is reduced if the model is not calibrated at CSD location? Does this mean that in case of assimilation of distributed soil moisture observations from remote sensing, within a distributed hydrological model, we would need to calibrate the model in each grid cell? I disagree with the author. The main purpose of data assimilation is to use real-time (noisy) observations to update the wrong estimate of the states of a dynamic model (not able to mimic the actual system states away from calibration point). Assimilation of observations at internal points of the catchment is very useful when model states are less accurate than real-time observations. If a model is able to correctly predict actual system states away from calibration points, why should someone bother to add complexity and uncertainty assimilating CSD observations? The literature provides many studies (e.g. Rakovec et al., 2012) in which hydrological models are updated using measurements at internal points, even if such observations are not used during model calibration.*

Thank you for this comment. I realize that I was not precise enough in that part of my Comment, which was improved in the revised version of the Comment.

To answer the key question in this Reviewer's comment (which I highlighted above), I remark that a model can predict wrong system states away from calibration points for different reasons (e.g., wrong/insufficient input data and/or poor calibration and/or structural model deficiencies). Assimilation of observations at internal points of the catchment can be extremely useful when model states are less accurate than real-time observations, but not when this lack of accuracy of model states is due to problems with model structure (or due to poor calibration of model parameters if such parameters are not updated through data assimilation along with the model states).

Therefore, I stress that the statements by M. Mazzoleni are reasonable, but they implicitly assume that the model structure (and the set of model parameter as well, since they were not updated through data assimilation in his work) is (are) able to correctly estimate, at the same time, both the internal states and the model outputs. Although this is a highly desirable feature for physical-based models (see also comment #3 of Reviewer #2), one must admit that it is not true in general and, reasonably, it is not true for the model application of the Bacchiglione River presented in Mazzoleni et al. (2017).

I try to clarify the question using first a hypothetical example. Consider a hydrological model, not calibrated at internal points, which provides the right prediction at the closing section as the result of wrong predictions at some internal points. The updating of model states at this internal points based on real data (i.e., different to the internal states needed to provide the 'correct' prediction at the outlet) will likely cause this model to produce worse predictions at the closing section with respect to no assimilation at all. This possible occurrence cannot be detected, nor assessed, if data to be assimilated are extracted from the model itself. Indeed, in this case synthetic data represent wrong internal states (with respect to reality), but they represent the best-fit scenario in terms of main model output.

The problem of assimilating data that are not coherent with internal model states (when this is due to poor estimation/identifiability of model parameters) could be limited by updating the model parameters along with the internal states of the model (as suggested by Reviewer #2), but this strategy could not be sufficient if the model has structural deficiencies.

As a practical example, consider the "Viale Diaz" floodplain, described in my Comment, which acts as a sort of in-line natural flood control reservoir on flood propagation. Since the attenuation of flood wave exerted by this floodplain can not be properly accounted for by the routing model used in Mazzoleni et al. (2017), the (hypothetical) assimilation of a correct flood hydrograph upstream of the Viale Diaz floodplain leads to incorrect streamflow predictions at Ponte degli Angeli, downstream of the "Viale Diaz" floodplain.

7. *I am puzzled with the sentence "Therefore, beside the key points identified by Mazzoleni et al. (2017), not only data, but also the model has to match specific requirements for data assimilation to be successful". What are these specific requirements that model has to match? Is the Author referring to the reliability of synthetic data at calibration points and to the capability of the model to represent truth states?*

The need to assimilate suitable crowdsourced data was assessed in Mazzoleni et al. (2017). With respect to the specific requirements that the model has to match, its ability of well representing the physics of the hydrological system (i.e., of correctly representing true internal states when forced by correct input data) is actually the key aspect. I tried to make this point clearer in the revised version of the Comment.

**3 RESPONSE TO THE COMMENTS OF REVIEWER #2**

*The author makes some significant critical remarks on the work of Mazzoleni et al. (2017) that are worth to be considered for publication.*

I thank Reviewer #2 for his/her appreciation of my Comment and for his/her very useful and precise suggestions, which are addressed in the following

1. *However, I would first advise to mellow the tone of the narrative.*

   Thanks for the suggestion. I revised the text of the paper to smooth the English and to fix many typos.

2. *In addition, I invite the author to make sure that the comments are more general and less focused on the upper Bacchiglione river catchment presented by Mazzoleni et al. (2017). In doing so, Section 2.1 should be reduced considerably, as most of the information and comments seem too specific, and might not be supported for the other test sites.*

   I thank the reviewer for his suggestion. In the revised version of the Comment, Section 2 was shortened and reorganized, trying to separate specific comments that refer to the upper Bacchiglione case study from general comments (Section 3). I agree that Sect. 2.1 is very specific. Nevertheless, I do believe that most part of this specificity is not meaningless for other test sites. Indeed, I remain convinced that much can be learned from in-depth analyses of specific cases. The opposite risk is the (often unperceived) oversimplification of real systems and processes in our schematic representations (i.e., models) of the reality.

   Besides its evident specificity, one of the goals of Sect. 2.1 is to highlight that real-world case studies are often far more complex than what can emerge from most of the applications reported in the literature (this is undoubtedly due to the need of limiting paper length). I am convinced that similarities with many other case studies can easily be found.

   Finally, the Comment is indeed a comment to a specific paper, and only one of the four case studies reported by Mazzoleni et al (2017) is here commented, since the contents of the Comment only apply to (semi-)distributed (and over-parameterized) models and to the assimilation of CSD data at location where the model cannot be calibrated. In the other test cases presented in Mazzoleni et al (2017), the Authors used a lumped model and CSD were assimilated only at calibration sections.

3. *The paper of Mazzoleni et al. (2017) aimed at investigating the value of information retained by crowdsourced data (CSD) when assimilated in surface flow models for flood prediction. Their work is admittedly a proof-of-concept study and the synthetic feature of CSD is quite clear, rather than "briefly mentioned". Their conclusions are correct so long as one assumes the model well represents the physics of the hydrological system, which is a fundamental hypothesis behind observation simulation system experiments. On the other hand, I agree that there seems to be an inherent tendency in Mazzoleni et al. (2017) to present results in a way that somehow overstates the importance of CSD.*

   I agree with the reviewer. The fact that a model well represents the physics of the hydrological system is a fundamental hypothesis for physically-based models, and is tacitly assumed in Mazzoleni et al. (2017). However, it must be stressed that this requirement is not necessarily matched when semi-distributed, physically based models are actually used as lumped models, i.e., they are calibrated only at the closing sections. Given the complexity of the Bacchiglione catchment, the relatively paucity of measured data, and the structure of the model used (see my answer to comment #6 of Reviewer #1 for further details), reasonably it is not true for the model application of the Bacchiglione River presented in Mazzoleni et al. (2017).

4. *There are, in my view, some major points that need to be highlighted: the method chosen for calibrating a model should be consistent regardless of the type of data used. For non-linear models, ensemble based data assimilation methods (e.g the EnKF or the PF) are attractive in that they can be used to update jointly model states and parameters and provide a direct measure of uncertainty. Note that these models cope directly with problems of over parameterization and equifinality since parameter posterior distributions are represented by ensembles. CSD can be instrumental to reduce model uncertainty. Indeed, one can assimilate these data together with traditional hydrologic observations, thereby reducing parameter uncertainty even in those regions where the original reliability of the model is inadequate. In general, the value of information of these data is strictly dependent on their quantity, quality, spatial-temporal distribution. Note that typical data assimilation algorithms are in principle able to screen out noisy data automatically, but need to be modified to tackle possible data bias, which otherwise leads to poorly calibrated models. Thus, it is important, regardless of the nature of the data, to verify if such bias exists before any data assimilation is applied.*

I thank the Reviewer for these interesting considerations. Ensemble based data assimilation methods are indeed powerful tools. On one hand, their use to jointly update model states and parameters can effectively circumvent the problem of uncertainty in model internal states at crowdsourced data points; on the other hand, such methods can help diagnosing poor identifiability of model parameters.

However, sophisticated tools to update jointly model parameters and states may fail if assimilating data at locations where the model is unable to correctly reproduce the actual physics of the system due to structural deficiencies. This occurrence is far from being rare in operational flood forecasting frameworks where (over)simplified models are commonly used in order to limit their computational burden.

While structural deficiencies can be a-priori conjectured through a close inspection of both the physical system and the model characteristics, it can be proved (and quantified) only by comparing model results with measured data (i.e., model validation). The "blind" use of CSD (i.e., its assimilation at locations where the model is neither calibrated nor verified) is at least questionable (see, e.g., the examples reported in the answer to comment #6 of Reviewer #1).

Finally, in the Reviewer's comment it is stressed the importance of detecting bias in data to be assimilated. This observation pertains also to the object of my Comment, since assimilating real (i.e., not synthetic) data at locations where the model is unable to reproduce the physic of the system is equivalent to assimilating biased data.

These considerations have been added to the revised version of the Comment.

**References**

[revised manuscript text omitted]

---

## Author Response (AR2)

**POINT-BY-POINT REPLY TO COMMENTS**

**1 RESPONSE TO THE EDITOR'S COMMENTS**

*The manuscript has now undergone 2 rounds of review. The first set of review comments were considered major and, therefore, the revised manuscript was sent out for review again after the author had made substantial changes to the manuscript.*

*In this round of reviews, Reviewer #1 notes that there has been significant improvement to the revised manuscript and recommends that the manuscript be accepted with minor revision. Reviewer #2 notes there are interesting points raised in this comment but felt that there was further major revision (and review) needed for the manuscript to be accepted.*

*In looking at both reviewers comments and my own review of the revised manuscript, the main points of this commentary still need clarification and be generalized enough to make the comment of interest to a wide audience. Reviewer #2 notes that this quite possible but that major revision is still needed.*

*To help the author clarify the contribution, the reviewers have noted their interpretations of the major points of this commentary and provide specific comments about how this can be done. I offer the same here to help with revisions: In my understanding of the contribution of this comment, I find of most interest the point that models are often called upon for applications that require information to make decisions of high societal value - such as flood forecasting - and that we have a responsibility as hydrologists to understand the limits and applicability of our models and the data that drive those models. As crowd-sourcing data (CSD) can now be used as a further tool to enhance hydrologic modeling efforts, out community needs to consider the use of CSD in our responsible assessment of the applicability of models to answer questions of such importance and relevance. I see the author using a recent paper on CSD, study area, and model to make this comment.*

*I believe that an additional revision effort, will continue to improve the general interest of the paper and elevate the impact of the comment. For this reason, I am recommending additional revision before the manuscript can be accepted.*

*I thank again the author and reviewers for the productive discussion and progress towards publication.*

Again, I am grateful to the Editor and the Reviewers for their suggestions, which I found insightful and helpful.

I used the Editor's suggestions mainly to improve the Introduction of the Comment, highlighting the importance of debating and deeply understanding both models and data characteristics in order to provide useful and reliable tools to support the decision-making related to flood forecasting and management. Now the beginning of the Introduction reads (additions in red):

> "Flood forecasting has a critical importance as it results in decisions of high societal value. It is essential to provide public authorities with the best combination of data and models in order to produce the most accurate flood predictions, and with a robust knowledge of the model behaviour in terms of reliability and uncertainty. Modellers thus have a responsibility to deeply assess the strengths and limitations of models, and to explore different kind of forcing data as well.
>
> Within this general picture, the topic of crowdsourced data is gaining increasing attention among hydrologists. Indeed, the availability of hydrometric data, collected by active citizens in the course of severe flood events, offers a new, unexpected chance to improve real-time flood forecasts. However, the use of crowdsourced data poses severe challenges to modellers since their information content, reliability, arrival frequency, and location are a-priori unknown (Mazzoleni et al., 2015, 2017; van Meerveld et al., 2017). In addition, long time series of crowdsourced data are in fact unavailable.
>
> In pioneering applications, crowdsourced data (CSD) collected in the upper part of a basin were assimilated into adaptive hydrological models..."

Moreover, I'm grateful to M. Mazzoleni as his suggestion helped in amending some imprecisions in the text and in better introducing the matter of the comment, and to the anonymous reviewer for giving a chance to make the manuscript significantly clearer and easily understandable.

**2 RESPONSE TO THE COMMENTS OF REVIEWER #1**

*The author clearly improved the manuscript and I really appreciate his effort. The structure of the comment is now clear and the main objective well described. I do believe that author critical remarks will be helpful for future publications in the same area.*

5     *Maurizio Mazzoleni*

I thank M. Mazzoleni (Reviewer #1) for his appreciation of the Comment and for his valuable comments and suggestions, which helped me to further improve my Comment.

1. *However, I still have a couple of comments/suggestions regarding section 3 (use of CSD in operation flood forecasting). I think this section needs more clarifications. In particular, from the abstract and introduction (page 2 lines 1-3) it seems*

10     *that the main objective of the comment is to discuss the effects of assimilation of unreliable synthetic CSD derived from a poorly calibrated model on the assimilation performances. Nonetheless, section 3 provides a qualitative analysis on the effect of model structural uncertainty in data assimilation, which is different than the original scope of this comment. The author need to clarify this issue or to better frame the scope of the comment.*

I agree. The two arguments mentioned by M. Mazzoleni are actually quite different and, although being mutually related

15     in the scope of the manuscript, they both deserve to be mentioned when introducing the main issues of the Comment. Accordingly, I modified the Abstract to read (additions in red):

"... In most real-world applications, hydrological models are calibrated using data from traditional sensors; CSD are typically collected at different locations, where (semi-)distributed models are not calibrated. As a result of either equifinality, poor model identifiability, and lacks in model structure, internal states of (semi-

20     )distributed models can hardly mimic the actual states of complex systems away from calibration points.  Synthetic CSD generated by such models are unreliable and do not allow to assess the effects of model structural uncertainty; their use may lead to overestimating the performance of CSD assimilation with respect to real applications. Additional guidelines are given that are useful for the a-priori evaluation of

25     crowdsourced data for real-time flood forecasting and, hopefully, to plan apt design strategies for both model calibration and collection of crowdsourced data. ..."

In the Introduction, the following modifications were made:

"A practical verification of the results by Mazzoleni et al. (2017) is indeed necessary; furthermore, particular attention has to be paid to  possible drawbacks inherent in the use of CSD for operational flood

30     forecasting and related to model structural uncertainty, which are not discussed in their proof-of-concept study."

2. *Page 5, lines 25-30: Is the author referring to the assimilation of synthetic or real CSD? I do agree that synthetic CSD estimated at different points than the calibration ones may not be accurate when model is poorly calibrated. This concept was already reported in section 2.3. On the other hand, I do not see the point of this paragraph if the author is referring*

35     *to real CSD. Obviously, a traditional physical sensor will provide more reliable observations than CSD if located at the same point. However, the benefit of CSD is in their spatial distribution and availability in points where physical sensors are not available, as stated already in the original paper. I suggest the author to clarify which type of CSD (synthetic or real) are considered in section 3.*

I agree that the paragraph at hand was not properly structured. I changed the paragraph by moving the first sentence at the

40     end of the paragraph, so that the reason why I put this paragraph at the beginning of section 3 should now be clear. In other words, since CSD typically do not refer to calibration points, we must look carefully at the model behaviour/performance away from calibration points. Now the paragraph at hand reads (moved text in green):

" In general, historical data recorded by traditional sensors are first used to calibrate a model; then, in real-time mode, the same sensors provide data both to force the model and to update the model states (e.g., Ercolani and Castelli, 2017); moreover, the reliability of data from traditional sensors outperforms that of CSD. Hence, from a practical point of view, CSD have limited usefulness at locations already equipped with traditional sensors. Since the natural purpose of CSD is to enhance (rather than replace) data from traditional sensors, CSD typically do not refer to model calibration points."

According to the reviewer's suggestion, I also added several references to "real" CSD in Section 3 (starting from the title of Section 3) in order to clarify what I was referring to.

3. *Page 6, lines 18-24: It is stated in the comment that data assimilation can be used only at calibration points in case of poorly calibrated model. I have some doubts about this statement (in particular page 6 lines 20-22). It is worth noting that even in case of not-properly calibrated model (which is the case of semi-distributed and distributed models) assimilation of reliable observations can help improving model performances. The author should include that a proper estimation of the model error, expressed by means of the error covariance matrix in linear DA or model spread in Ensemble method, is necessary to ensure an appropriate assimilation process in case of poor model calibration. There are many studies in which observations are assimilated at internal points different than the calibration points. This is the case of assimilation of remote sensing observations in case of distributed model calibrated only in few discharge locations (when available). I suggest the author to improve this part.*

I remark that the sentence referred to in this comment is not valid in general; as stated in the text, it specifically applies to cases in which "only internal states are updated". Anyway, I added a sentence to this paragraphs, which now reads (additions in red):

"... assimilation of CSD in operational flood forecasting can be helpful provided that the model is able to well represent the physics of the system at locations where CSD are collected. Of course, data assimilation can contribute, in many cases, to improve such a representation. However, when only internal states are updated (as in Mazzoleni et al., 2017), this condition is met if (and only if) the model is properly calibrated and verified at locations where CSD refer to. Otherwise, correcting internal states of a poorly calibrated model can even lead, in principle, to worse predictions at the outlet than performing no corrections at all (Crow and Van Loon, 2006). It is undoubtedly difficult to assess this issue when only synthetic CSD, generated by the same model, are available for testing the overall method."

4. *Page 7, lines 28-30: I do not agree with this statement. It is because of the random behaviour and involvement of citizens that CSD location cannot be determined a-priori. CSD can be provided at any point of the basin/river and not only at calibration ones. There has been many studies in which mobile applications are developed to correctly estimate river velocity or flow (e.g. Lüthi et al. 2014) by using river cross section (which can be assumed rectangular). Definitely, such tools will provide an uncertainty estimation of the flow characteristics of the river which may lead to a wring update of the model states at interior points. The hope is that more reliable tools for accurately measuring river flow will be developed in the next years.*

I agree, the sentence was formulated in a too much general terms. I then restricted the sentence to water level CSD, which actually need the existence of a rating curve to be converted into streamflow data.

"It must be observed that, while scarce control on the collection of CSD can be exerted during significant flood events, the locations at which citizens can collect CSD of water levels is always determined a-priori, since the availability of rating curves is a necessary condition in order to convert water levels into discharges. The amount of measured data needed to develop reliable rating curves can also be profitably used to calibrate the model at those sections as well."

5. *I have a final question to the author. Since CSD can be assimilated only at points when model is calibrated and physical sensors (more reliable than CSD) are already installed at these points, is the author implicitly suggesting that CSD from citizens should not be used for improving flood predictions even in case that system states are well represented?*

No, I'm not saying that. CSD can be assimilated at locations different from calibration points, but in this case attention must be paid to update not only model internal states, but also model parameters. Moreover, the suitability of model structure in correctly represent the physics of the real system at these points (when the model is properly forced, of course) must be a-priori checked.

6. *I hope these suggestions will help the author to further improve his valuable contribute to the use of CSD from citizens for improving flood prediction.*

I thank again M. Mazzoleni for his helpful suggestions.

**3   RESPONSE TO THE COMMENTS OF REVIEWER #2**

*Thanks for the opportunity to review the revised version o this comment. I read both this version and the preceding discussion with interest. Honestly, I have somewhat split feelings about this comment. On the one hand some valuable points are made, but on the other hand the 'comment-aspects' are not fully clear.*

I thank Reviewer #2 for his/her effort in reviewing the Comment and in providing useful suggestions, which are addressed in the following

1. *As I see it, the author makes two main comments: 1) the use of synthetic data for the crowdsourced observations and 2) the model choice and application. While the author is very critical about the first point, I would disagree to phrase this as general as done in the comment. Studies using synthetic data can actually be quite informative to investigate the question on how valuable such data potentially could be if they would be available. I would argue this is a suitable approach can actually provide guidance on how to collect crowed sourced data (see also van Meerveled et al, in review, HESS-D, as an example of this approach). The author needs to provide more convincing arguments why the approach in general is not suitable or on where exactly he sees short-comings of the particular implementation of this approach in the study by Mazzoleni et al.*

I admit I was quite puzzled in reading this comment. I carefully re-read the manuscript and I did not find anything, with reference to the use of synthetic data, saying that "the approach in general is not suitable". I searched for all the occurrences of the word "synthetic" throughout the text, and I found that synthetic data (and limitations related to their use) are always mentioned with reference to the use of (semi-)distributed (and overparametrized) hydrological models and, more specifically, with reference to the Bacchiglione case study presented by Mazzoleni et al. (2017). I agree with the reviewer that, in general, the use of synthetic data can actually be quite informative. As a matter of fact, I remark that Mazzoleni et al. (2017) used synthetic data in three additional case studies and, in my Comment, I was not critical at all about those applications. However, synthetic data must be reliable or, alternatively, their uncertainty/inaccuracy have to be (fairly) known. In my comment, I show that this is not the case when synthetic data are generated away from calibration points by a (semi-)distributed hydrological model. Specifically, if synthetic data are extracted from the best-fit scenario and than assimilated into the same model, they are obviously leading to better performance than real crowdsourced data. In other words, the question on how valuable such data potentially could be if they would be available can not be properly answered if synthetic data are surely better (but no one can say how much better) than real crowdsourced data.

I then concluded that maybe the Abstract and the Introduction were not well formulated, and thus formed a sort of wrong perception of what I was commenting on in the following sections of the manuscript.

Accordingly, I revised the Abstract to read (additions in bold):

"In their recent contribution, Mazzoleni et al. (2017) investigated the integration of crowdsourced data (CSD) in hydrological models to improve the accuracy of real-time flood forecasts. The Authors used synthetic CSD (i.e., not actually measured), because real crowdsourced data were not available at the moment of the study. In their work, which is actually a proof-of-concept study, Mazzoleni et al. (2017) showed that assimilation of CSD improves the overall model performance; the impact of irregular frequency of available CSD, and that of data uncertainty, were also deeply assessed. However,  the use of synthetic CSD in conjunction with a semi-distributed hydrological model deserves further discussion. In most real-world applications, hydrological models are calibrated using data from traditional sensors; CSD are typically collected at different locations, where (semi-)distributed models are not calibrated. As a result of either equifinality, poor model identifiability, and lacks in model structure, internal states of (semi-)distributed models can hardly mimic the actual states of complex systems away from calibration points.  Synthetic CSD generated by such models are unreliable and do not allow to assess the effects of model structural uncertainty; their use may lead to overestimating the performance of CSD assimilation with respect to real applications. Additional guidelines are given that are useful for the a-priori evaluation of crowdsourced data for real-time flood forecasting and, hopefully, to plan apt design strategies for both model calibration and collection of crowdsourced data."

I also found an overstatement in the Introduction, and I amended the text to read (additions in red):

"A practical verification of the results by Mazzoleni et al. (2017) is indeed necessary; furthermore, particular attention has to be paid to  possible drawbacks inherent in the use of CSD for operational flood forecasting and related to model structural uncertainty, which are not discussed in their proof-of-concept study."

Finally, once clarified that in my Comment it was never stated that the approach is in general not suitable, I remark that Section 2.3 is completely devoted to explain where exactly I see shortcomings in the particular implementation of this approach in the study by Mazzoleni et al. (2017), in a way that M. Mazzoleni found this comment reasonable and useful. Nonetheless, to further clarify this key point, I substantially enhanced Section 2.3 (see my answer to your next comment).

2. *The second point I actually find more interesting. The author nicely provides reasons on why a so called physical model is not as physical as one might think. I find this discussion really helpful, although it could be a bit more to the point. I can clearly sympathize with the argument that the use of such a 'physical but still not so physical' model has implementations for the result in a study which heavily relies on model calibration, and probably the use of a simpler model by Mazzoleni et al. could have been appropriate. However, it is not clear from the comment in which way the author thinks this could have influenced the results.*

I realized that, in the previous version of the manuscript, this part was not clear enough. Therefore, I substantially enhanced Section 2.3, which is entirely devoted to explain this point. Now it reads (additions in red, moved text in green):

"In the Bacchiglione case study, Mazzoleni et al. (2017) calibrated the model using measured rainfall data to well reproduce the streamflow hydrograph at the closing section (call this post-event simulation "scenario 1"). Then they forced the model with predicted rainfall fields that were completely different from the actual storm event ("scenario 2"); in this case, the discharge simulated using forecasted input was very different from that obtained using recorded rainfall, with a significant time shift and errors in predicted discharge ranging between 25 and 50% at the flood peak (and up to 90% if considering synchronous data).  In the "scenario 3", similarly to

the "observing system simulation experiment" (OSSE) approach, synthetic streamflow CSD extracted from the "scenario 1" were assimilated into a new run using the same forcing as in the "scenario 2".  Not surprisingly, the model performance in the "scenario 3" was significantly better than in the "scenario 2".

The Authors claimed that  synthetic CSD they used are realistic.  For this condition to be met, given that these CSD are results of the model itself, the model must represent well the physics of the real system (i.e., it must be calibrated or, at least, verified) at locations where CSD are first generated and then assimilated; this is a fundamental hypothesis behind the OSSE approach. As a matter of fact, the synthetic CSD used in Mazzoleni et al. (2017) for the Bacchiglione case study are representative of the model internal states of the best-fit scenario.  However, recalling that such CSD do not refer to model control points, nothing can actually be said about the model performance at locations where CSD are generated and, as a consequence, about their accuracy. Real CSD are then expected to be farther from the best-fit scenario than the synthetic CSD generated by the model; that is, real CSD are likely biased with respect to the synthetic CSD actually used. Therefore, assimilation of real crowdsourced data can not be as effective as that performed in Mazzoleni et al. (2017).

From one point of view, such an inconsistency could have led Mazzoleni et al. (2017) to overrate the importance of CSD , as they considered issues related to CSD precision, but not accuracy (Mazzoleni et al., 2016).  From a more general point of view, additional care must be taken in operational flood forecasting when assimilating CSD into (semi-)distributed hydrological models at locations other than model control points. This last point is further discussed in the next section."

3. *From the comment it is clear that certain aspects of Mazzoleni et al. could have done differently or have been described clearer. However, to be really useful a comment has to be specific and raise issues of general importance. As argued above, I do not agree with the fundamental critic against the use of synthetic data. While point 2 still could be of general interest, it is not yet formulated in such a way. In the current form it mainly describes the details but misses to frame this in a more general discussion on which model to use when, including the consequences of using a too complex model. To summarize, while the comment raises an important point which could be of general interest, some significant work is needed to make the comment as useful as it could be.*

I agree with the reviewer that, to be really useful, a comment has to respond to two different (and substantially opposite) needs: it has to analyse and debate specific aspects, and has to raise issues of general importance as well.

I hope that, with the enhancements made in the revised version of the paper (as described above), the part referring to specific aspects is now sufficiently clear and complete. I am also convinced that the issue concerning the critic against the use of synthetic data (not really present in the manuscript) is now clarified.

On the other hand, I point out that a "general discussion on which model to use when" goes far beyond the scope of the present manuscript that, being in fact a specific commentary on the paper by Mazzoleni et al. (2017), mainly deals with possible shortcomings in the assimilation of crowdsourced data into (semi-)distributed hydrological models for real-time flood forecasting.

I believe that the comment, at least for people operating in the field of flood forecasting and interested in the use of crowd-sourced data, already presents issues of quite general interest. Nevertheless, in Section 3, which is mainly devoted to generalize the main specific issues of the Comment, I added some sentences in order to make the reasoning more clear. Specifically, in the revised version of the manuscript, it is explicitly stated that CSD are typically spatially distributed. Accordingly, spatially explicit models are needed in order to take advantage from this kind of data. Unfortunately, physically-based, (semi-)distributed models suffer from equifinality, poor identifiability of model parameters, and structural deficiencies, leading to possible shortcomings related to the assimilation of CSD referring to locations different

from the calibration point of the model. Finally, guidelines and possible solutions (along with associated limitations) are discussed.

**References**

Lüthi, B., Philippe, T. and Peña-Haro, S.: Mobile device app for small open-channel flow measurement, International Environmental Modelling and Software Society (iEMSs) 7th Intl. Congress on Env. Modelling and Software, San Diego, CA, USA, 2014.

[revised manuscript text omitted]

---

## Author Response (AR3)

**1 RESPONSE TO THE EDITOR'S COMMENTS**

*The reviewer of the revised comment has determined that the manuscript is a worthwhile contribution to the literature and is ready for publication. I have now read the revised comment, and while I agree, the comment is a worthwhile contribution, there are still areas of the comment that require editorial revisions before the manuscript can be accepted.*

*I now ask the author to address these comments. I will assess that the comments have been addressed before final acceptance of the manuscript for publication.*

- *Please proofread thoroughly the entire manuscript.* Done. Many typos were amended throughout the text.

- *Abstract: The abstract does not mention flood forecasting nor does it seem to capture the main points of the comment. I would look to the summary of the comment, particularly the last 2 paragraphs, which contains some nice text about the main points of the comment.* Thanks for the suggestion. I revised the Abstract for it to capture the main points of the comment, by removing unnecessary information and adding more general comments.

- *p. 1, line 3: Change "moment" to "time" and delete the word "actually".* Done.

- *p. 1, line 10: You do not use the word "unreliable" later to describe the use of synthestic CSD. I would remove this sentence here and replace with some of the points made in the summary.* This sentence has been removed and replaced with a more general comment.

- *p. 1, lines 15-16: Move the phrase "in order to produce the most accurate flood predictions" to the start of the sentence.* Done.

- *p. 1, line 18: Delete ", and to explore different kind of" and "as well".* Done.

- *Introduction, paragraphs 1 and 2: Abbreviate CSD after the first use of crowdsourced data (line 19) and then use CSD after that (line 21).* Done.

- *p.1, line 20: Change "unexpected" to "exciting".* Done.

- *p. 2, line 1: Move the reference to Mazzoleni to after the phrase "In pioneering applications".* Done.

- *p. 2, line 3: Change to read "In this recent work".* Done.

- *p. 4, line 27: Change "hard" to "difficult".* Done.

- *p. 5, line 14: What is meant by "closing section" here? Rephrase using a hydrology term.* Sorry for the wrong expression. I've changed it with "basin outlet" in different parts of the text.

- *p. 5, line 18, Change to read "In their...".* Done.

- *p. 5, line 20: Provide the reason for why this was not surprising. This will help reiterate the point you are making in this paragraph.*

  Now the sentence reads (addition in red):

  "Not surprisingly, the model performance in the "scenario 3" was significantly better than in the "scenario 2", as the synthetic CSD they assimilated were representative of the model internal states in the best-fit scenario."

- *p. 5, line 22: Change to read "...the model must well represent the physics...".* Done.

- *p. 5, line 24: I do not see where OSSE has been spelled out before it is abbreviated here.* "OSSE" is spelled out in the preceding paragraph.

– *p. 5, lines 25-29: These sentences are still a bit unclear. This text should be more clear in describing one of the central points of your comment. Provide more detail so that the reader is able to easily grasp the point.*

Thank you for the suggestion. The paragraph has been reformulated to read (additions in red):

" The synthetic CSD used in Mazzoleni et al. (2017) for the Bacchiglione case study are  drawn from the model internal states  under best-fit  conditions. Thus, when the model is forced with different (wrong) input data, their assimilation is expected to be as successful as possible in updating the model states toward the best-fit scenario. However,  the accuracy of such synthetic CSD is questionable, since they do not refer to model control points (i.e., they are drawn from the semi-distributed model at locations where the model is neither calibrated nor verified), so nothing can actually be said about the model performance at these locations  In a sense, synthetic CSD used by Mazzoleni et al. (2017) are *optimal* (in view of assimilation performance) rather than *realistic*. Since  real CSD are likely biased with respect to the synthetic CSD actually used assimilation of real  CSD can not be as effective as that performed in Mazzoleni et al. (2017)."

– *p. 5, line 33: Delete the last sentence of this paragraph.* Done.

– *Section 3 is nicely written.* Thanks!

– *p. 6, line 3: Change to read: "However, this comment points out that attention must be paid...".* Done.

– *p. 6, line 4: Please add a reference to some literature that use ensemble based methods for flood forecasting.*

I added references to two literature works that use ensemble based methods just after where ensemble methods are first mentioned in the text (it is at p. 7). Now the sentence reads (addition in red):

"As an alternative for operational forecasting, ensemble based data assimilation methods (e.g., the Ensemble Kalman Filter or the Particle Filter) can be used to update jointly model states and parameters and to provide a direct measure of uncertainty (Moradkhani et al., 2005; Salamon and Feyen, 2009)."

– *p. 8, line 5: Change to CSD where "crowdsourced data" is used.* I changed "crowdsourced data" to CSD everywhere in the text, but not here, where "crowdsourced" is used as opposed to "traditional".

– *Wording needs to be softened in several areas of the comment:*

*p. 1, line 21: Delete "severe".* Done.

*p. 1, line 23: Delete "in fact" and expand on why a lack long time series of CSD data are a further limitation.* Done. Now the sentence reads (addition in red): "In addition, long time series of (CSD) are unavailable, thus complicating efforts to assess their effectiveness in improving flood prediction."

*p. 2, line 16: Change to read "..., which affects the interpretation of the usefulness of CSD.".* Done.

*p. 5, line 21: Change "claimed" to "argued".* Done.

*p. 5, line 24: Delete "As a matter of fact...".* Done.

*p. 5, line 30: Change to read "It is possible that such an inconsistency....".* Done.

*p. 5, line 31-32: Change to read "Therefore, additional care must be taken...".* Done.

*p. 7, line 6: Change to read "As an alternative for...".* Done.

*p. 7, line 27-28: Change to read "and flow routing processes face limitations...".* Done.

[revised manuscript text omitted]